# Prognostic Efficacy of Complete Blood Count Indices for Assessing the Presence and the Progression of Myxomatous Mitral Valve Disease in Dogs

**DOI:** 10.3390/ani13182821

**Published:** 2023-09-05

**Authors:** Min-Jung Jung, Jung-Hyun Kim

**Affiliations:** Department of Veterinary Internal Medicine, College of Veterinary Medicine, Konkuk University, #120 Neungdong-ro, Gwangjin-gu, Seoul 05029, Republic of Korea; ryu03106@konkuk.ac.kr

**Keywords:** complete blood count, neutrophil-to-lymphocyte ratio, monocyte-to-lymphocyte ratio, platelets-to-lymphocyte ratio, myxomatous mitral valve disease, pulmonary edema

## Abstract

**Simple Summary:**

The neutrophil-to-lymphocyte ratio (NLR), monocyte-to-lymphocyte ratio (MLR), and platelet-to-lymphocyte ratio (PLR) are inflammatory indicators calculated using blood counts that serve as potential biomarkers in various diseases. This study investigated healthy dogs and those with a common heart disease, myxomatous mitral valve disease (MMVD). Dogs with MMVD had higher NLR, MLR, and PLR, which could be useful for diagnosing and predicting this condition.

**Abstract:**

This study investigated the complete blood count (CBC) indices, including neutrophil-to-lymphocyte ratio (NLR), monocyte-to-lymphocyte ratio (MLR), and platelet-to-lymphocyte ratio (PLR) in dogs with myxomatous mitral valve disease (MMVD) and analyzed their correlation with conventional biomarkers, as well as the effect of CBC indices on survival time in dogs with MMVD. Medical records of 75 healthy controls and 249 dogs with MMVD from March 2015 to October 2022 were analyzed. The NLR, MLR, and PLR were calculated by dividing the absolute counts of the CBC parameters. Dogs with MMVD had significantly higher NLR, MLR, and PLR than healthy dogs (all *p* < 0.0001), especially those in the symptomatic MMVD group with pulmonary edema (*p* < 0.0001, *p* = 0.0002, and *p* = 0.0387, respectively). The NLR and MLR were significantly correlated with N-terminal pro-B type natriuretic peptide levels (both *p* < 0.0001). The CBC indices showed potential as biomarkers for detecting the presence of MMVD (all *p* < 0.0001) and severity of MMVD (*p* < 0.0001, *p* < 0.0001, and *p* = 0.006, respectively) using receiver operating characteristic curve analysis. The negative effects of increased NLR, MLR, and PLR on survival were confirmed using Kaplan–Meier curve analysis. In conclusion, NLR, MLR, and PLR could be cost-effective and readily available potential diagnostic and prognostic biomarkers for MMVD in dogs.

## 1. Introduction

Myxomatous mitral valve disease (MMVD) is the most common cause of congestive heart failure (CHF) and the most common acquired cardiac disease in dogs, accounting for 75% of all heart disease cases [1,2,3]. Conventional echocardiography and cardiac biomarkers, such as N-terminal pro-B-type natriuretic peptide (NT-proBNP) and cardiac troponin I (cTnI), are the gold standard methods for diagnosing CHF [1]. However, as these procedures require specialized equipment and well-trained operators and are relatively costly, more accessible biomarkers for predicting MMVD and monitoring its progression are needed to facilitate cardiac disease management. Identifying reliable inflammatory biomarkers in dogs with heart failure has received considerable attention recently since inflammation and oxidative stress have been found to be involved in the pathogenesis of cardiovascular diseases [4,5,6].

The neutrophil-to-lymphocyte ratio (NLR), monocyte-to-lymphocyte ratio (MLR), and platelet-to-lymphocyte ratio (PLR) are important systemic inflammatory indicators that can be easily calculated from complete blood count (CBC) tests. These CBC indices help comprehensively evaluate the inflammatory response compared to individual inflammatory cells because they consider the interrelationship between different cell types, thus assessing the immune and inflammatory status more accurately [7,8]. Therefore, these indices serve as biomarkers of systemic inflammatory responses and are potential diagnostic and prognostic biomarkers in human patients with inflammatory and neoplastic conditions, as well as cardiovascular diseases [9,10,11,12]. Indeed, increased NLR and MLR are associated with cardiovascular diseases, such as coronary artery disease, stroke, mitral valve disease, and adverse cardiac events [9,13,14]. Furthermore, PLR is a relatively novel marker of inflammation and thrombosis and has been used as a prognostic biomarker for cardiovascular diseases [7,15]. Recently, in veterinary medicine, the efficacy of the NLR as a prognostic factor has been demonstrated in cats with hypertrophic cardiomyopathy [16]. Although leukocyte and platelet parameters have been investigated in dogs with inflammatory or neoplastic diseases [17,18,19], there have only been a few studies in dogs with spontaneous cardiac disease [20,21], and little is known about their relationships with conventional heart failure biomarkers and echocardiographic variables [22]. Furthermore, while CBC indices such as NLR, MLR, and PLR have been demonstrated to be potential biomarkers of the inflammatory response in inflammatory diseases, such as sepsis or pancreatitis and neoplastic diseases, such as oropharyngeal tumors in dogs [17,23], their evaluation in cardiac diseases has rarely been reported.

Therefore, this study aimed to investigate whether NLR, MLR, and PLR increase depending on the presence and severity of MMVD and to analyze their correlation with conventional biomarkers or echocardiographic parameters. This study also aimed to investigate the effects of CBC indices on survival time.

## 2. Materials and Methods

### 2.1. Animals

The medical records of all canine MMVD cases recorded at the University of Konkuk Veterinary Medicine Teaching Hospital between March 2015 and October 2022 were electronically reviewed for suitability. The inclusion criteria for the study were hematology performed within 24 h of the echocardiogram and echocardiographic evidence of MMVD with or without clinical signs such as dyspnea, coughing, exercise intolerance, or collapse. Based on physical examination and echocardiography, healthy dogs with no history or clinical evidence of underlying cardiac diseases were included as controls. The exclusion criteria were non-MMVD cardiac disease, hematologic disease, severe liver or renal dysfunction, autoimmune disease, or acute or chronic infectious disease. Dogs with periodontal disease or clinical signs related to inflammation, as well as those receiving medications that are known to alter white blood cells (WBC) (e.g., corticosteroids) in the previous three months, were also excluded.

This study was approved by the Institutional Animal Care and Use Committee of Konkuk University (Approval No.: KU23057).

### 2.2. Classification of Myxomatous Mitral Valve Disease

MMVD diagnosis was established by identifying typical mitral valve lesions, such as thickened and/or prolapsing mitral valve leaflets, and by confirming mitral regurgitation through color Doppler echocardiography. The MMVD group was further categorized based on the classification system of the American College of Veterinary Internal Medicine (ACVIM) [1].

### 2.3. Laboratory Analysis

Blood samples were collected via jugular venipuncture into ethylenediaminetetraacetic acid (EDTA)-containing tubes (BD Biosciences, Franklin Lakes, NJ, USA) to measure hematologic parameters and the conventional cardiac biomarker, NT-proBNP. The CBC and automated differential counts were determined within 1 h after collecting the blood samples using a ProCyte Hematology Analyzer (IDEXX Laboratories, Inc., Westbrook, ME, USA), which provided total WBC, neutrophils, monocytes, platelets, and lymphocyte counts/mL. Baseline NLR, MLR, and PLR were calculated by dividing the absolute counts of these parameters. Samples for EDTA plasma were centrifuged at 1000× *g* at 4 °C for 15 min, and the plasma was transferred to a commercially available laboratory (IDEXX Laboratories, Seoul, Republic of Korea) and analyzed for NT-proBNP concentration (pmol/L) using an IDEXX ELISA (IDEXX Laboratories, Leipzig, Germany).

### 2.4. Thoracic Radiography

Right lateral and ventrodorsal views of thoracic radiographs during maximal inspiration were obtained to evaluate heart size by vertebral heart score (VHS) and to identify abnormalities in the lung field to determine the presence of pulmonary edema (PE) using radiographic equipment (Titan 2000V^®^, Comed Medical System, Gyenggi-do, Republic of Korea). The VHS was calculated by measuring the long and short axes, and their measurements were compared with the length of the thoracic vertebrae starting from the cranial edge of the 4th thoracic vertebra. The long axis represents the distance from the carina to the cardiac apex, and the short axis represents the maximum diameter perpendicular to the long axis of the heart.

### 2.5. Echocardiography

Echocardiography was performed and evaluated by an experienced investigator (JH-K) to diagnose MMVD, assess its severity, and exclude other cardiac diseases. Echocardiography and analysis were performed by an experienced investigator using an ultrasound unit (HD 15, Philips Ultrasound, Bothell, WA, USA) equipped with 3.0–8.5 MHz phased-array transducers in all dogs. All echocardiographic examinations were conducted using standard imaging planes, according to the recommendations of the American Society of Echocardiography [24], and by using two-dimensional, M-mode, color flow Doppler and tissue Doppler. The two-dimensional variables measured included LV internal dimensions at end-diastole (LVIDd) and end-systole (LVIDs). LV fractional shortening (FS) was calculated using the following formula: LV FS = [LVIDd − LVIDs]/LVIDd × 100%. Left atrial (LA) size was assessed using standard right parasternal long axis and short axis views. The variables measured included LA diameter (LASAX) and aortic diameter (Ao) measured from a right parasternal short axis view in early diastole. The LASAX to Ao ratio (LA/Ao) was then calculated. Transmitral velocities were recorded using pulsed-wave Doppler, and the measured variables included the peak velocity of early diastolic transmitral flow (E) and the peak velocity of late diastolic transmitral flow (A). The ratio of peak E to peak A (E/A) was then calculated. Tissue Doppler imaging (TDI) was performed with the highest available transducer frequency to record the velocity of the lateral mitral annular motion from the left apical parasternal long axis view. The following variables were measured: peak early diastolic velocity (E′), peak late diastolic velocity (A′), and peak systolic velocity (S′).

### 2.6. Statistical Analysis

Statistical analyses were performed using commercial statistical software (IBM SPSS Statistics, version 25.0, IBM Corp., Armonk, NY, USA; GraphPad Prism, version 8, GraphPad Software, Inc., San Diego, CA, USA; R, version 4.04, R Development Core Team, Vienna, Austria). The Shapiro–Wilk test was applied to test for the normality of distribution, and continuous variables were defined as mean ± standard deviation when normally distributed or median and interquartile range (IQR) when the distribution was non-normal. Based on the normality of the results, Student’s *t*-test was performed to compare continuous variables between the control and MMVD groups. Propensity score matching (PSM) was performed based on age. The propensity score was generated using a logistic regression model, and a 1:3 nearest-neighbor matching method was performed. The standardized difference in the means was less than 10% after matching (caliper = 0.01). One-way analysis of variance (ANOVA) followed by Tukey’s multiple comparison test was used to compare the control and each stage of the MMVD groups. Additionally, the Kruskal–Wallis test, followed by the Mann–Whitney test with Bonferroni’s correction for multiple comparisons, was used to compare NT-proBNP among the control and each stage of MMVD groups. The Mann–Whitney test was used to compare the differences in CBC ratios between the symptomatic groups with and without pulmonary edema. Pearson correlation and linear regression analyses were used to assess the correlations between CBC indices and conventional biomarkers and radiographic and echocardiographic measurements. Receiver operating characteristic (ROC) curve analyses were performed to verify the diagnostic accuracy of CBC indices in predicting the presence and severity of MMVD. The cut-off was chosen according to the highest of the various combinations of specificity and sensitivity using Youden’s index (Y = sensitivity + specificity − 1). Survival analysis was performed for dogs that reached cardiac death (primary endpoint), and all dogs that were alive, dead from non-cardiac disease, or lost to follow-up were right censored. The CBC indices were subclassified by tertiles and analyzed using Kaplan–Meier curves, and the differences were compared using the log-rank test. Statistical significance was set at *p* < 0.05.

## 3. Results

### 3.1. Characteristics of the Study Population

A total of 324 dogs (19 intact males, 155 castrated males, 39 intact females, and 111 spayed females) with a median age of 10.98 years (10.98 ± 4.14) were enrolled in this study. Of these, 75 dogs were healthy and considered as controls, while 249 were affected by MMVD. According to the ACVIM classification, 92 dogs were classified as stage B1, 49 as stage B2, 92 as stage C, and 18 as stage D.

The most represented breeds for dogs with MMVD were Maltese (n = 102; 40.96%), Poodle (n = 48; 19.28%), Shitzu (n = 41; 16.47%), Pomeranian (n = 27; 10.84%), mixed breed (n = 20; 8.03%), Yorkshire terrier (n = 15; 6.02%), and Bichon Frise and Chihuahua (10 each; 4.02%). The other breeds were represented by fewer than nine dogs each. The most representative breeds of normal dogs were Maltese (n = 18; 24%), Poodle (n = 14; 18.67%), Pomeranian (n = 10; 13.33%), mixed breed (n = 8; 10.67%), Bichon Frise (n = 7; 9.33%), and Yorkshire terrier (n = 3; 4%). The other breeds were each represented by < 3 dogs. There were no statistically significant differences in systolic blood pressure (*p* = 0.300) or heart rate between the groups (*p* = 0.370). The VHS and LA/Ao ratio, indicators of cardiac size remodeling, showed a significant increase starting from MMVD stage B2. Additionally, the VHS and LA/Ao ratios in MMVD C were significantly different from those in MMVD B2 (*p* = 0.010 and *p* = 0.003, respectively) and MMVD D (*p* = 0.038 and *p* < 0.0001, respectively). Clinical, radiographic, and echocardiographic variables of the dogs are presented in Table 1.

The conventional biomarker, NT-proBNP, was measured in 127 dogs, and the median value of NT-proBNP for each group is shown in Table 2.

### 3.2. Comparison of Clinical and Hematological Variables between Healthy Dogs and Those with MMVD

A comparison of the clinical and hematological variables between the MMVD and healthy groups is shown in Table 3.

Significant differences in age, leukocyte, neutrophil, monocyte, lymphocyte, NLR, MLR, and PLR were observed between the healthy and MMVD groups. To avoid an imbalance caused by a significant age difference between the healthy and MMVD groups, PSM was used to match participants with similar age scores to avoid bias. As a result, significant differences in neutrophil (control, 5.61 ± 1.97; MMVD, 8.68 ± 4.40; *p* < 0.0001), monocyte (control, 0.45 ± 0.30; MMVD, 0.73 ± 0.44; *p* < 0.0001), lymphocyte (control, 2.22 ± 0.74; MMVD, 1.86 ± 0.55; *p* = 0.007), and platelet (control, 331.67 ± 128.51; MMVD, 456.34 ± 161.89; *p* = 0.002) counts were shown. The NLR (control, 2.67 ± 0.87; MMVD, 4.83 ± 2.50; *p* < 0.0001), MLR (control, 0.22 ± 0.13; MMVD, 0.40 ± 0.24; *p* < 0.0001), and PLR (control, 169.82 ± 82.68; MMVD, 265.72 ± 122.22; *p* < 0.0001) in dogs with MMVD were significantly higher than those in the healthy group, and these indices exhibited a greater difference in mean values compared to the individual values of each parameter (Figure 1).

### 3.3. Comparison of Clinical and Hematological Variables between Healthy Dogs and Those with MMVD at Different Stages

MMVD cases were further categorized according to the ACVIM classification system standard [1] as follows: MMVD B1, n = 90; MMVD B2, n = 49; MMVD C, n = 92; MMVD D, n = 18. A comparison of clinical and hematological variables between healthy dogs and those with MMVD at different stages is shown in Table 4. There were significant differences in age between the control group and each stage of MMVD (*p* < 0.0001); however, there was no significant difference between the MMVD stages (MMVD B1 vs. MMVD B2, *p* = 0.991; MMVD B2 vs. MMVD C, *p* = 0.994; MMVD C vs. MMVD D, *p* = 0.985, respectively). Neutrophil counts showed a significant difference between MMVD B2 (6.73 ± 1.76) and the control (5.46 ± 1.74, *p* = 0.001), while leukocyte and monocyte counts showed significant differences between MMVD C (14.58 ± 5.66 and 0.99 ± 0.52, respectively) and the control (9.61 ± 5.35, *p* < 0.0001 and 0.48 ± 0.29, *p* < 0.0001, respectively). All three values were significantly higher than the normal range in MMVD C; however, in MMVD D, leukocyte and neutrophil counts increased above the normal range (17.07 ± 5.19, *p* < 0.0001 and 13.70 ± 4.82, *p* < 0.0001, respectively). Compared to the control, lymphocyte and platelet counts showed significant differences in MMVD B1 (1.85 ± 0.71, *p* < 0.0001 and 400.83 ± 128.56, *p* = 0.001, respectively), MMVD B2 (2.00 ± 0.64, *p* < 0.0001 and 405.39 ± 111.41, *p* = 0.001, respectively), MMVD C (1.99 ± 0.60, *p* < 0.0001 and 485.78 ± 152.10, *p* = 0.001, respectively), and MMVD D (1.74 ± 0.53, *p* = 0.036 and 504.56 ± 188.81, *p* < 0.0001, respectively).

Compared to the control group, the NLR was significantly higher in MMVD B1 (3.23 ± 0.98, *p* = 0.0032), MMVD B2 (3.56 ± 1.08, *p* = 0.0004), MMVD C (5.71 ± 2.18, *p* < 0.0001), and MMVD D (8.38 ± 3.22, *p* < 0.0001). Additionally, the NLR in MMVD C was significantly different from those in MMVD B2 (*p* < 0.0001) and MMVD D (*p* < 0.0001) groups. The MLR values in MMVD B1 (0.28 ± 0.14, *p* = 0.0311), MMVD B2 (0.33 ± 0.23, *p* = 0.0018), MMVD C (0.51 ± 0.23, *p* < 0.0001), and MMVD D (0.74 ± 0.30, *p* < 0.0001) were significantly higher than those in the control group (0.20 ± 0.11). Additionally, MLR in MMVD C was significantly different from those in MMVD B2 (*p* < 0.0001) and MMVD D (*p* < 0.0001) groups. The PLR in MMVD B1 (234.27 ± 92.46, *p* < 0.0001), MMVD B2 (218.32 ± 81.44, *p* < 0.0001), MMVD C (261.67 ± 104.07, *p* < 0.0001), and MMVD D (326.77 ± 177.01, *p* < 0.0001) were significantly higher than those in the control group (0.20 ± 0.11). Additionally, the PLR in MMVD C was not significantly different from those in MMVD B2 (*p* = 0.681) and MMVD (*p* = 0.082) groups; however, the PLR in MMVD D was significantly higher than that in MMVD B2 (*p* = 0.0007) (Figure 2).

### 3.4. Comparison between CBC Indices According to the Presence of PE in the Symptomatic MMVD Groups

The Mann–Whitney test was performed to assess the impact of pulmonary edema (PE) on inflammation levels in the symptomatic MMVD groups (MMVD C and MMVD D) using the NLR, MLR, and PLR. Out of the 110 patients with clinical symptoms, 25 had PE, while 85 patients did not. The results showed that in the presence of PE, there were significant increases in NLR (4.75 [IQR: 3.73–6.43] vs. 6.98 [5.41–8.80], *p* < 0.0001), MLR (0.39 [IQR: 0.29–0.51] vs. 0.56 [0.42–0.67], *p* = 0.0002), and PLR (239.10 [IQR: 165.66–318.35] vs. 327.44 [202.95–408.84], *p* = 0.0387) (Figure 3).

### 3.5. Correlations of CBC Indices with Conventional Biomarker, Radiographic, and Echocardiographic Variables

A correlation analysis performed using a Pearson correlation analysis revealed that the NLR and MLR were significantly correlated, moderately and positively, with the conventional biomarker NT-proBNP (r = 0.493; *p* < 0.0001 and r = 0.404; *p* < 0.0001, respectively). PLR was significantly but weakly correlated with NT-proBNP levels (r = 0.285; *p* < 0.0001). NLR and MLR showed a significant but weakly positive correlation with VHS (r = 0.382; *p* < 0.0001 and r = 0.323; *p* < 0.0001, respectively), LA/Ao ratio (r = 0.396; *p* < 0.0001 and r = 0.351; *p* < 0.0001, respectively), and FS (r = 0.261; *p* < 0.0001 and r = 0.261; *p* < 0.0001, respectively) (Figure 4) and showed a very weak correlation with E/E′ (r = 0.189; *p* = 0.001 and r = 0.180; *p* = 0.002, respectively); however, no correlation was found with other selected echocardiographic variables (EF, *p* = 0.762 and *p* = 0.438; E/A, *p* = 0.677 and *p* = 0.970; E′/A′, *p* = 0.300 and *p* = 0.600, respectively). PLR was not significantly correlated with radiographic or echocardiographic variables (VHS, *p* = 0.110; EF, *p* = 0.859; E/A, *p* = 0.835; E′/A′, *p* = 0.389; E/E,’ *p* = 0.391). The correlation coefficient was interpreted using general guidelines on the coefficient extent [25]: very weak correlation, 0 < r ≤ 0.19; weak correlation, 0.2 ≤ r ≤ 0.39; moderate correlation, 0.4 ≤ r ≤ 0.59; high correlation, 0.6 ≤ r ≤ 0.79; very high correlation, 0.8 ≤ r ≤ 1.

### 3.6. Efficacy of CBC Indices for Predicting MMVD

ROC curve analysis was used to assess the efficacy of CBC indices in predicting MMVD after PSM. With a cut-off level of 3.73, NLR predicted MMVD with a sensitivity of 61.46% and specificity of 92.50% (AUC, 0.826; 95% confidence interval [CI]: 0.757–0.895), *p* < 0.0001). The best cut-off value of MLR for predicting the presence of MMVD was 0.30, with 59.38% sensitivity and 87.50% specificity (AUC, 0.780, 95% CI: 0.699–0.862, *p* < 0.0001). Moreover, the best PLR cut-off level for predicting MMVD was 206.25, with 68.75% sensitivity and 75.00% specificity (AUC, 0.760, 95% CI: 0.673–0.848, *p* < 0.0001) (Figure 5).

### 3.7. Diagnostic Efficacy of CBC Indices for Detecting the Severity of MMVD

The ROC curve was used to analyze the efficacy of CBC indices in detecting the severity of MMVD by classifying the mild-to-moderate group (MMVD B1 and B2) and the advanced group (MMVD C and D). The cut-off point of 3.97 for NLR predicted advanced MMVD with a sensitivity of 88.18% and specificity of 81.30% (AUC, 0.890, 95% CI: 0.848–0.931, *p* < 0.0001). For MLR, the cut-off point of 0.38 for MLR predicted advanced MMVD with a sensitivity of 74.55% and specificity of 79.14% (AUC,0.825, 95% CI: 0.773–0.878, *p* < 0.0001). Moreover, the cut-off point of 243.59 for PLR predicted advanced MMVD with a sensitivity of 55.45% and specificity of 64.75% (AUC, 0.602, 95% CI: 0.531–0.673, *p* = 0.006) (Figure 6).

### 3.8. Survival Analysis According to the Increase in the NLR, MLR, and PLR

The survival time of MMVD cases was significantly associated with an increase in the NLR, MLR, and PLR (Figure 7). Each CBC index was classified into tertiles for survival analysis. The mean follow-up time for the MMVD group was 179 weeks (95% CI: 145.50–212.82). Thirty-one dogs died or were euthanized due to heart failure. The median survival time (MST) in dogs with high NLR (NLR>4.88) was 56 weeks (95% CI: 38.15–72.85), showing a significantly shorter survival time than that in dogs with both moderate NLR (3.45 < NLR < 4.88) and low NLR (NLR < 3.45) (*p* = 0.0006; *p* < 0.0001, respectively). The MST in dogs with high MLR (MLR > 0.45) was 67 weeks (95% CI: 42.46–91.54), which was significantly shorter than that in dogs with moderate MLR (0.26 < MLR < 0.45) and low MLR (MLR < 0.26) (*p* < 0.0001 and *p* = 0.0178, respectively). Furthermore, the MST in dogs with high PLR (PLR > 280.11) was 80 weeks (95% CI: 36.40–123.60), showing a significantly shorter survival time than that in dogs with moderate PLR (195.36 < PLR < 280.11) and low PLR (PLR < 195.36) (*p* = 0.0089; *p* = 0.0231, respectively).

## 4. Discussion

In total, 324 dogs were enrolled in this retrospective study. This study demonstrated that CBC indices were significantly higher in dogs with MMVD than in healthy dogs, especially in the advanced-stage group. The CBC indices, especially NLR and MLR, were proven to be correlated with the conventional biomarker, NT-proBNP, together with VHS, LA/Ao ratio, and FS, and showed efficacy in predicting MMVD and detecting its severity. Moreover, Kaplan–Meier curve analysis confirmed that high levels of NLR, MLR, and PLR had a negative effect on survival time.

It has been recently discovered that heart failure syndrome encompasses changes in the inflammatory and immune systems, in addition to hemodynamic and neurohormonal changes [26,27]. Inflammation not only contributes to myocardial dysfunction but also to adverse outcomes such as endothelial dysfunction and cardiac cachexia [5]. The pro-inflammatory roles of neutrophils, monocytes, lymphocytes, and platelets in cardiovascular diseases are well known [28]. Neutrophils secrete inflammatory mediators that can degenerate vascular walls [9]. Although they may not directly contribute to systolic dysfunction, they may indicate other inflammatory mediators that are directly involved in the pathogenesis of cardiac dysfunction. The interaction between inflammatory cells and neutrophils within the myocardium occurs via the release of lysosomal enzymes and arachidonic acid metabolites, causing myocardial dysfunction [29]. Additionally, activated neutrophils may induce monocyte recruitment to inflammation sites by releasing various cytokines [8]. Monocytes have been found to predict mortality in cardiovascular diseases and are involved in adverse remodeling, which may lead to heart failure [14,30,31]. Monocytosis is associated with angiogenesis secondary to myocardial damage and increased left-sided filling pressure in patients with CHF [21]. An increase in monocyte-derived endothelial progenitor cells in patients with CHF supports this finding [21]. Furthermore, lymphocytes are important immune cells in cardiovascular disease [31]. The main cause of the decrease in lymphocyte counts is unclear, but it is probably related to increased corticosteroid levels or other neurohormonal alterations due to heart failure [27]. In humans, relative lymphopenia is independently related to survival time and is considered a prognostic indicator in patients with heart failure [32]. Markedly lower total lymphocyte counts and percentages of lymphocyte subpopulations have also been observed more frequently in dogs with severe CHF than in the controls [5]. Platelets release pro-inflammatory mediators, and activated platelets play an important role in cardiovascular diseases by stimulating thrombus formation resulting from the rupture of endothelial cells [7,33]. Shear stress from mitral regurgitation has been proposed to alter platelet function and lifespan, contributing to MMVD progression in dogs [34].

Compared to these individual blood parameters, NLR, MLR, and PLR are relatively less susceptible to several factors, such as age, sex, dehydration, and blood specimen handling, allowing a more accurate assessment of the degree of inflammation and immune response status [7,8]. Moreover, because these CBC indices are derived from the analysis of various types of cells through routine blood examinations, they are easily measurable and more stable than other inflammatory markers or cytokines [35]. Therefore, CBC indices have been thoroughly studied in human medicine over the last few years to understand their role in cardiovascular diseases and have been demonstrated to be associated with CHF [29,36,37,38,39]. In veterinary medicine, many studies on CBC indices as inflammatory biomarkers have also been conducted in inflammatory and neoplastic diseases [17,18,23], and very recently, the values of these indices have been investigated in dogs and cats with heart disease [16,40].

In this study, high neutrophil, monocyte, and platelet counts and low lymphocyte counts within the reference range were found in dogs with MMVD compared with healthy controls, which led to significantly increased NLR, MLR, and PLR in dogs with MMVD. To minimize the effect of age differences on hematological markers between the control and MMVD groups, PSM was conducted. In a previous study on dogs with MMVD, significant differences were found in NLR and MLR between the control and MMVD groups, and the mean values of NLR and MLR in cases with MMVD were 3.48 and 0.32, respectively [40]. In contrast, in the present study, the mean values of NLR and MLR were higher, precisely 4.83 and 0.40, respectively. These differences are believed to be due to the larger number of cases with MMVD in the present study, particularly with MMVD C and MMVD D, which was approximately five times higher than that in the previous study. Moreover, contrary to previous research, which showed no significant difference in PLR between the healthy and MMVD groups [40], the present study found a significant increase in PLR in the MMVD group compared to that in the control group. These results are consistent with those of previous veterinary studies that found an increase in platelets and a decrease in lymphocytes in the MMVD group compared to healthy controls [2,27,41] and with previous human studies that confirmed the significance of PLR as a prognostic factor in patients with heart failure [29,38].

We then compared the hematological variables between the control group and the groups with different stages of MMVD. Lymphocyte and platelet counts were significantly different between the MMVD B1 and control groups, whereas WBC, neutrophil, and monocyte counts showed a significant increase in MMVD C and MMVD D compared with MMVD B1, MMVD B2, and healthy dogs. Although the median value of each parameter in MMVD C did not exceed the upper value of the reference range, leukocytosis and neutrophilia were found in MMVD D, suggesting that systemic inflammation may be present in the advanced stage of MMVD. The mechanisms by which leukocytosis affects CHF progression include leukocyte aggregation, proteolytic impairment, microvascular occlusion, and impaired revascularization [4]. In previous studies, WBC, neutrophil, and monocyte counts in the International Small Animal Cardiac Health Council (ISACHC) III group were significantly higher than those in ISAHC-II, ISACHC-I, and healthy dogs, which is in accordance with the results of this study [4,20,27,42]. The CBC indices, NLR, MLR, and PLR showed a significant increase in MMVD B1 compared with the control. No significant differences were observed between MMVD B1 and B2. However, NLR and MLR were significantly different from MMVD C, and PLR was significantly different from MMVD D. This result suggests that inflammatory status increases as the disease progresses. Several previous studies showing that increased levels of NLR, MLR, and PLR are associated with increased inflammation in various cardiac diseases, such as coronary arterial disease and mitral valvular disease, support this finding [28,37,43]. Consequently, the CBC indices showed stronger statistical power in predicting the presence and severity of MMVD compared to each type of cell analyzed individually, demonstrating their potential as prognostic markers in dogs with MMVD. In addition, while the previous study grouped MMVD C and MMVD D together for classification [40], the present study analyzed these two groups separately, which was possible by securing a large sample size, thereby providing more distinct evidence of the differences between MMVD stages.

Pulmonary edema, the most severe manifestation of CHF, occurs as a result of a sudden increase in pulmonary capillary hydrostatic pressure [44]. The disruption of both the capillary endothelium and alveolar epithelium, along with the accumulation of edema fluid, leads to an elevation in inflammatory products such as neutrophils and biomarkers of oxidative stress [44,45]. In this study, we found that the NLR, MLR, and PLR were significantly higher in symptomatic groups with pulmonary edema compared to those without pulmonary edema. These findings suggest the presence of acute inflammation in the alveoli, which may contribute to further tissue damage. The association between pulmonary edema, a manifestation of CHF, and increased systemic inflammation highlights the role of inflammation in dogs with MMVD.

The correlation between the CBC indices and NT-proBNP, a cardiac neurohormone released in response to increased wall stretching of the left ventricle and myocardial ischemia [10], was evaluated for the first time in dogs. NLR and MLR showed moderate positive correlations, and PLR showed a weak positive correlation with NT-proBNP. In human medicine, a study has reported that NLR and MLR show a higher positive correlation with NT-proBNP than PLR [46], which is similar to the results of this study. This suggests that CBC indices, especially the NLR and PLR, may play important roles in cardiac remodeling and myocardial damage. Moreover, similar to previous studies in dogs [40], NLR and MLR showed a weak positive correlation with VHS and LA/Ao ratio. However, in addition to this, significant correlations of NLR and MLR with FS were identified, which were not confirmed in a previous study. This suggests that NLR and MLR may be associated with structural changes and systolic function and may play important roles in the pathophysiology of canine MMVD. A study in humans, which claimed that the positive correlation between NLR and FS (r = 0.388, *p* = 0.023) suggests an association between inflammation and myocardial function, supports this finding [47]. However, no significant correlation was found between the CBC indices and echocardiographic variables of diastolic function (E/A, E′/A′, and E/E′).

ROC analysis was performed after PSM between the control and MMVD groups to investigate the value of the CBC indices as predictors of MMVD. Based on the AUC, the NLR (AUC, 0.826) showed very good accuracy at the cut-off of 3.73, and the MLR (AUC, 0.780) and PLR (AUC, 0.760) demonstrated good accuracy at the cut-offs of 0.30 and 206.25, respectively. The CBC indices, especially NLR and MLR, showed high specificities of 92.50% and 87.50%, respectively; however, since all three indices showed low sensitivity of less than 70%, there could be a possibility of misdiagnosing cases with MMVD as healthy when using these values alone. Therefore, it is expected that CBC indices could be used more effectively in detecting MMVD while minimizing false positives when considered together with existing diagnostic methods or biomarkers. To evaluate the possibility of using CBC indices as indicators for predicting the severity of MMVD, ROC analysis was conducted in the mild-to-moderate (MMVD B1 and MMVD B2) and advanced groups (MMVD C and MMVD D). The NLR (AUC, 0.890) and MLR (AUC, 0.825) showed very good diagnostic values for detecting heart failure at the cut-offs of 3.97 and 0.38, respectively, while PLR (0.602) showed insufficient diagnostic value at the cut-off of 243.59. In particular, NLR and MLR had high sensitivities of 88.18% and 74.55%, respectively, and high specificities of 81.30% and 79.14%, respectively. The NLR cut-off was similar to that of the NLR cut-off of 3.96, which showed adverse cardiac events in a previous human study [48]. Therefore, CBC indices could be used as useful indicators for predicting the presence and severity of MMVD, particularly NLR and MLR, which showed high efficacy in detecting disease progression.

Research on CBC indices and MMVD in dogs has been limited. However, a recent study provided insights into the association between NLR, MLR, and MMVD in dogs [40]. In the previous study, NLR and MLR were found to predict MMVD severity in 128 dogs. In our present study, we expanded the sample size to 324 dogs and further classified advanced MMVD groups (MMVD C and D) into subgroups, allowing for more comprehensive categorization. This enabled us to assess not only the diagnostic and prognostic value of NLR and MLR but also that of PLR. Additionally, we investigated the relationship between the occurrence of PE and these indices, evaluating their potential as prognostic factors.

This is the first report to evaluate whether increased NLR, MLR, and PLR impact survival time in dogs with MMVD. The negative effects of increased NLR, MLR, and PLR were confirmed by Kaplan–Meier analysis. MMVD cases with high NLR (NLR > 4.88), high MLR (MLR > 0.45), and high PLR (PLR > 280.11) showed a significant reduction in MST compared to those in the low and moderate groups for each index. The MST for the high NLR, high MLR, and high PLR groups were 56, 67, and 80 weeks, respectively. Similar findings have been reported in human medicine, demonstrating that elevated NLR, MLR, and PLR are associated with poorer outcomes in patients with cardiovascular diseases and long-term mortality [13,37,39]. Recently, in veterinary medicine, it has been confirmed that cats with hypertrophic cardiomyopathy and a high NLR have a shorter survival time; moreover, NLR has been proven to be an independent prognostic indicator [16]. In addition to the findings of previous studies, this study provides the first evidence that NLR, MLR, and PLR have predictive and prognostic values for mortality in dogs with MMVD.

There is a need for easily accessible and cost-efficient biomarkers, especially for predicting MMVD severity. Although multiple prediction tests are available, clinicians often prefer simplified options, particularly in emergencies. NLR, MLR, and PLR, which can be derived from a routine CBC test, are readily accessible and convenient biomarkers that may have predictive value in MMVD dogs. These ratios may be particularly useful in settings where other more specific or costly biomarkers are unavailable.

The major limitation of this study was that potential biases may exist due to the retrospective nature of this study. To minimize this limitation, PSM was performed; however, the possibility of bias cannot be completely excluded. In addition, other inflammatory markers, such as C-reactive protein (CRP) or circulating inflammatory cytokines, were not evaluated in this study because of limited resources. Most dogs underwent cardiac examination and did not undergo CRP measurements unless they exhibited leukocytosis. Among the 324 dogs, only 22 showed leukocytosis. CRP was measured in only eight dogs. Owing to the lack of these data, they could not be included in the analysis. Furthermore, spot CBC indices were used rather than follow-up values, while serial values and comparative analyses would have been more informative. Therefore, further large-scale randomized follow-up studies are needed to evaluate the independent predictive value of the CBC indices.

## 5. Conclusions

This study demonstrated that the NLR, MLR, and PLR were higher in dogs with MMVD, and as the disease progressed, the ratios further increased. Additionally, the presence of PE in symptomatic MMVD dogs was linked to even higher levels of these inflammatory ratios. The CBC indices, particularly NLR and MLR, were significantly correlated with conventional biomarkers, and the efficacy of NLR and MLR in detecting the presence and the severity of MMVD was confirmed. In addition, higher NLR, MLR, and PLR were associated with shorter survival times. Therefore, CBC indices can be used as cost-effective and readily available potential diagnostic and prognostic biomarkers for MMVD in dogs. Moreover, these indices can be particularly valuable in situations where conventional diagnostic tests are pending due to financial constraints, patient compliance issues, or situations of emergency.

## Figures and Tables

**Figure 1 animals-13-02821-f001:**
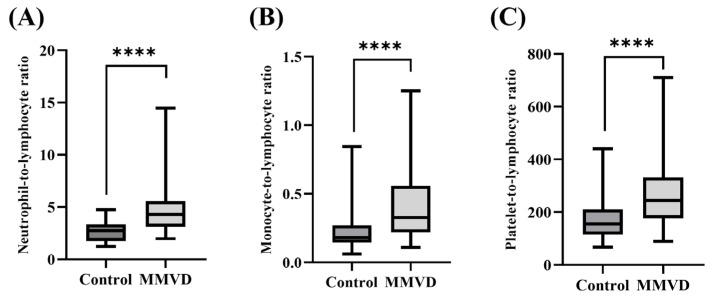
Comparison of neutrophil-to-lymphocyte ratio, monocyte-to-lymphocyte ratio, and platelet-to-lymphocyte ratio between healthy dogs and those with myxomatous mitral valve disease after propensity score matching. The (**A**) NLR, (**B**) MLR, and (**C**) PLR are found to be significantly higher in dogs with MMVD than in the control group. (*p* < 0.0001, *p* < 0.0001, *p* < 0.0001, respectively). The asterisk indicates a statistically significant difference. (**** *p* < 0.0001). NLR, neutrophil-to-lymphocyte ratio; MLR, monocyte-to-lymphocyte ratio; PLR, platelet-to-lymphocyte ratio; MMVD, myxomatous mitral valve disease.

**Figure 2 animals-13-02821-f002:**
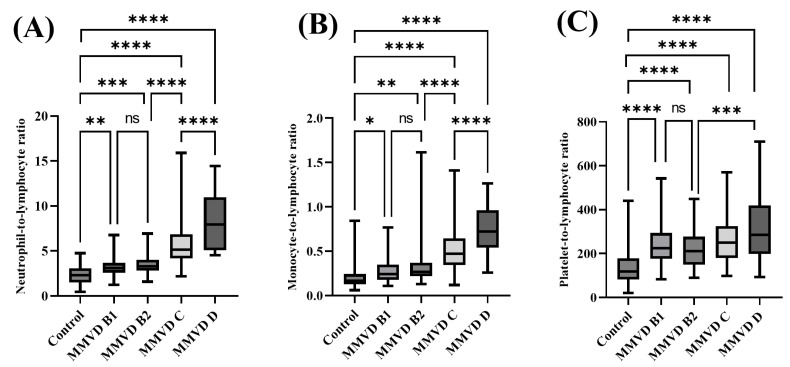
Comparison of neutrophil-to-lymphocyte ratio, monocyte-to-lymphocyte ratio, and platelet-to-lymphocyte ratio between healthy dogs and those with various stages of myxomatous mitral valve disease. The (**A**) NLR, (**B**) MLR, and (**C**) PLR were significantly higher in all MMVD stages than those in the control. The asterisk indicates a statistically significant difference. (**** *p* < 0.0001; *** *p* range, 0.0001–0.0020; ** *p* range, 0.0021–0.0331; * *p* range, 0.0332–0.05; ns, not significant). NLR, neutrophil-to-lymphocyte ratio; MLR, monocyte-to-lymphocyte ratio; PLR, platelet-to-lymphocyte ratio; MMVD, myxomatous mitral valve disease.

**Figure 3 animals-13-02821-f003:**
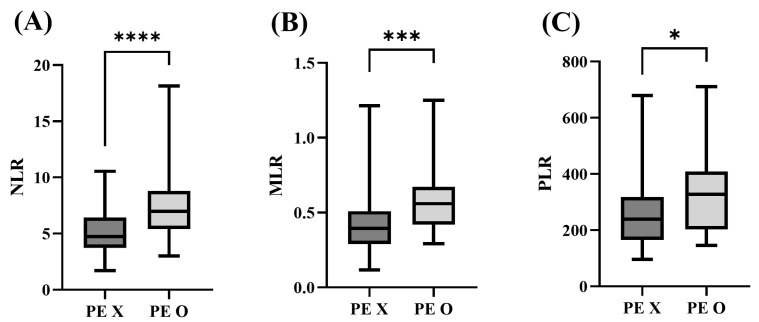
Comparison between NLR, MLR, and PLR of symptomatic groups with and without pulmonary edema. The (**A**) NLR, (**B**) MLR, and (**C**) PLR of symptomatic groups with pulmonary edema were significantly higher compared to the group without pulmonary edema. (*p* < 0.0001, *p* = 0.0002, and *p* = 0.0387, respectively). The asterisk indicates a statistically significant difference. (**** *p* < 0.0001; *** *p* range, 0.0001–0.0020; * *p* range, 0.0332–0.05). MLR, monocyte-to-lymphocyte ratio; NLR, neutrophil-to-lymphocyte ratio; PE, pulmonary edema PLR, platelet-to-lymphocyte ratio.

**Figure 4 animals-13-02821-f004:**
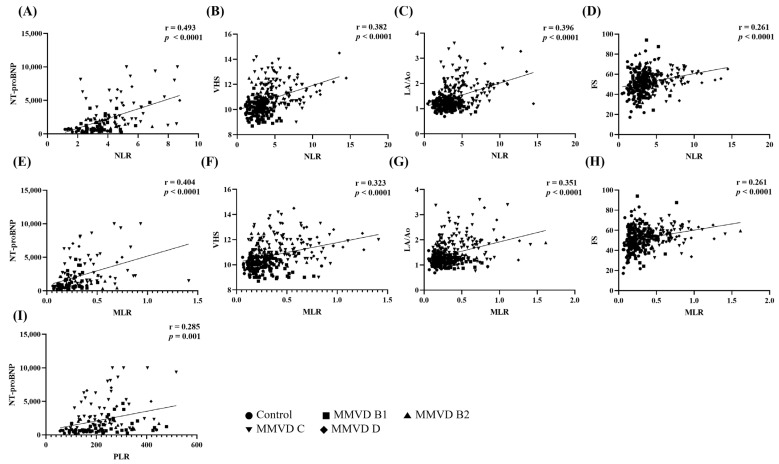
Correlations of neutrophil-to-lymphocyte ratio, monocyte-to-lymphocyte ratio, and platelet-to-lymphocyte ratio with conventional biomarker, radiographic, and echocardiographic variables. (**A**–**D**) Correlation of NLR between NT-proBNP, VHS, LA/Ao ratio, and FS; (**E**–**H**) Correlation of MLR between NT-proBNP, VHS, LA/Ao ratio, and FS; (**I**) Correlation between PLR and NT-proBNP. The NLR and MLR showed a moderate correlation with NT-proBNP (*p* < 0.0001 and *p* < 0.0001, respectively), and the PLR showed a weak correlation with NT-proBNP (*p* = 0.001). The NLR and MLR showed a weak correlation with VHS, LA/Ao ratio, and FS. NLR, neutrophil-to-lymphocyte ratio; MLR, monocyte-to-lymphocyte ratio; PLR, platelet-to-lymphocyte ratio; NT-proBNP, N-terminal pro-B type natriuretic peptide; VHS, vertebral heart score; LA/Ao, left atrium to aorta ratio; FS, fractional shortening.

**Figure 5 animals-13-02821-f005:**
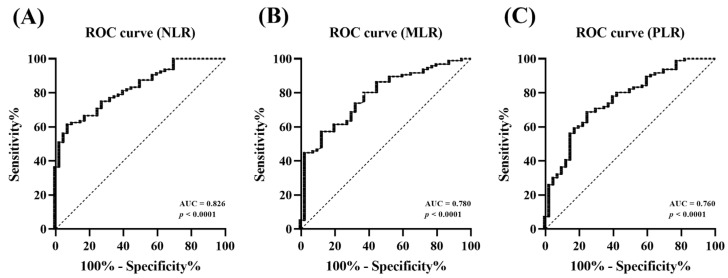
Receiver operating characteristic curve of complete blood count indices for predicting myxomatous mitral valve disease after propensity score matching. The AUCs are (**A**) 0.826 for NLR (*p* < 0.0001), (**B**) 0.780 for MLR (*p* < 0.0001), and (**C**) 0.760 for PLR (*p* < 0.0001) for predicting MMVD. AUC, area under the curve; NLR, neutrophil-to-lymphocyte ratio; MLR, monocyte-to-lymphocyte ratio; PLR, platelet-to-lymphocyte ratio; MMVD, myxomatous mitral valve disease.

**Figure 6 animals-13-02821-f006:**
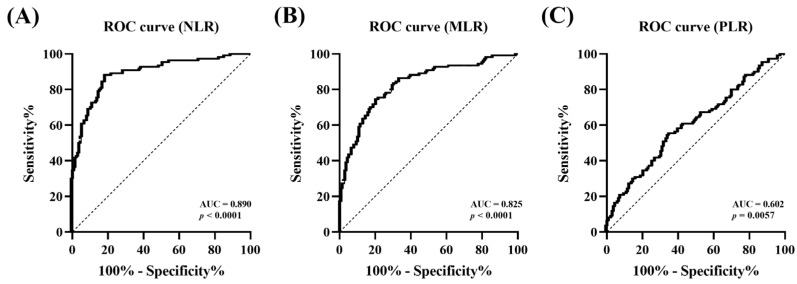
Receiver operating curve analysis of the diagnostic accuracy of complete blood count indices for predicting the severity of myxomatous mitral valve disease. The AUCs were (**A**) 0.890 for NLR (*p* < 0.0001), (**B**) 0.825 for MLR (*p* < 0.0001), and (**C**) 0.602 for PLR (*p* = 0.006) for predicting the severity of MMVD. AUC, area under the curve; NLR, neutrophil-to-lymphocyte ratio; MLR, monocyte-to-lymphocyte ratio; PLR, platelet-to-lymphocyte ratio; MMVD, myxomatous mitral valve disease.

**Figure 7 animals-13-02821-f007:**
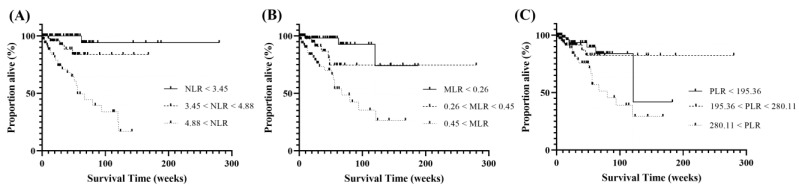
Kaplan–Meier survival curves showing the effect of increased neutrophil-to-lymphocyte ratio, monocyte-to-lymphocyte ratio, and platelet-to-lymphocyte ratio on survival in dogs with myxomatous mitral valve disease. (**A**) High NLR (NLR > 4.88), (**B**) high MLR (MLR > 0.45), and (**C**) high PLR (PLR > 280.11) groups show a significant decrease in median survival time compared to those in moderate and low groups (*p* = 0.0006 and *p* = 0.0001, *p* = 0.0001 and *p* = 0.0178, *p* = 0.0089, and *p* = 0.0231, respectively). NLR, neutrophil-to-lymphocyte ratio; MLR, monocyte-to-lymphocyte ratio; PLR, platelet-to-lymphocyte ratio.

**Table 1 animals-13-02821-t001:** Characteristics of healthy dogs and those with various stages of myxomatous mitral valve disease.

Group	Control	MMVD B1	MMVD B2	MMVD C	MMVD D
Number of dogs	75	90	49	92	18
Clinical variables					
Sex (male/female)	40/35	52/38	29/20	45/47	8/10
[IM:CM]/[IF:SF]	[6:34]/[17:18]	[6:46]/[6:32]	[3:26]/[3:17]	[3:42]/[9:38]	[1:7]/[4:6]
Age (years)	6.48 ± 4.46	12.21 ± 3.07 ^a^	12.35 ± 2.29 ^a^	12.51 ± 2.95 ^a^	12.06 ± 3.40 ^a^
SBP (mmHg)	135.21 ± 14.82	141.39 ± 14.95	138.84 ± 11.87	136.98 ± 15.02	134.89 ± 17.08
Heart rate (beats/min)	140.20 ± 28.00	144.30 ± 27.89	142.52 ± 27.32	149.39 ± 29.57	159.78 ± 17.64
Radiographic variable					
VHS	10.01 ± 0.50	9.90 ± 1.17	10.98 ± 0.70 ^a,b^	11.54 ± 1.03 ^a,b,c^	12.28 ± 1.05 ^a,b,c,d^
Echocardiographic variables					
FS (%)	45.76 ± 12.83	52.34 ± 10.62	54.46 ± 10.04 ^a^	56.16 ± 8.35^a^	54.54 ± 8.11 ^a^
EF (%)	76.00 ± 15.10	92.54 ± 83.23	85.59 ± 7.72	86.60 ± 7.80	85.56 ± 8.20
LA/Ao	1.16 ± 0.23	1.19 ± 0.50	1.45 ± 0.29 ^a,b^	1.83 ± 0.59 ^a,b,c^	2.14 ± 0.65 ^a,b,c,d^
E/A	2.13 ± 8.85	0.92 ± 0.26	0.98 ± 0.26	1.32 ± 0.68 ^b,c^	1.86 ± 1.07
E′/A′	1.11 ± 1.59	0.71 ± 0.24	0.79 ± 0.38	1.07 ± 0.53	1.25 ± 0.78
E/E′	11.50 ± 3.38	12.59 ± 2.92	13.21 ± 4.10	12.60 ± 5.28	14.69 ± 9.77

All *p*-values are obtained using ANOVA, and all continuous parameters are reported as mean ± standard deviation. ^a^ The value is significantly (*p* < 0.05) different compared to that of the control group; ^b^ The value is significantly (*p* < 0.05) different compared to that of the MMVD B1 group; ^c^ The value is significantly (*p* < 0.05) different compared to the of the MMVD B2 group; ^d^ The value is significantly (*p* < 0.05) different compared to that of the MMVD C group. MMVD, myxomatous mitral valve disease; IM, intact male; CM, castrated male; IF, intact female; SF, spayed female; SBP, systolic blood pressure; VHS, vertebral heart score; FS, fractional shortening; EF, ejection fraction; LA/AO, left atrium to aorta ratio; MV E/A, transmitral flow E wave velocity to A wave velocity ratio; Emax, maximal systolic elastance; MV E′/A′, E′ wave velocity to A′ wave velocity ratio; MV E/E,’ E wave velocity to E′ wave velocity ratio.

**Table 2 animals-13-02821-t002:** Conventional biomarker levels of healthy dogs and those with various stages of myxomatous mitral valve disease.

Group	Control	MMVD B1	MMVD B2	MMVD C	MMVD D
Number of dogs	22	38	19	43	5
Conventional biomarker					
NT-proBNP (pmol/L)	668.50 (446.50–738.00)	748.50 (574.50–1574.50)	1458.00 (883.00–2126.00) ^a^	2745.00 (1820.00–5515.00) ^a,b,c^	5000.00 (3177.00–6842.00) ^a,b,c^

Data are presented as median (interquartile range) values. The Mann–Whitney test with Bonferroni’s correction was used. Statistical significance is set at *p* < 0.005. ^a^ The value is significantly (*p* < 0.005) different than that in the control group; ^b^ The value is significantly (*p* < 0.005) different than that in the MMVD B1 group; ^c^ The value is significantly (*p* < 0.005) different than that in the MMVD B2 group. NT-proBNP, N-terminal pro-B type natriuretic peptide.

**Table 3 animals-13-02821-t003:** Comparison of clinical and hematological variables between healthy dogs and those with myxomatous mitral valve disease.

Variables	Before PSM			After PSM		
	Control Group (n = 75)	MMVD Group (n = 249)	*p*-Value	Control Group (n = 40)	MMVD Group (n = 96)	*p*-Value
Clinical variables						
Age (years)	6.48 ± 4.45	12.34 ± 2.91 ^a^	<0.0001	9.75 ± 3.40	10.73 ± 3.01	0.099
Hematological variables						
WBC (K/μL)	9.61 ± 5.35	11.56 ± 5.22 ^a^	0.006	9.79 ± 7.09	11.74 ± 4.96	0.069
Neutrophil (K/μL)	5.46 ± 1.74	8.47 ± 4.49 ^a^	<0.0001	5.61 ± 1.97	8.68 ± 4.40 ^a^	<0.0001
Monocyte (K/μL)	0.48 ± 0.29	0.77 ± 0.55 ^a^	<0.0001	0.45 ± 0.30	0.73 ± 0.44 ^a^	<0.0001
Lymphocyte (K/μL)	2.43 ± 1.05	1.92 ± 0.65 ^a^	<0.0001	2.22 ± 0.74	1.86 ± 0.55 ^a^	0.007
Platelet (K/μL)	311.58 ± 114.18	440.61 ± 145.39 ^a^	<0.0001	331.67 ± 128.51	456.34 ± 161.89 ^a^	0.002
NLR	2.33 ±0.93	4.58 ± 2.33 ^a^	<0.0001	2.67 ± 0.87	4.83 ± 2.50 ^a^	<0.0001
MLR	0.20 ± 0.11	0.41 ± 0.25 ^a^	<0.0001	0.22 ± 0.13	0.40 ± 0.24 ^a^	<0.0001
PLR	137.61 ± 77.59	247.94 ± 106.00 ^a^	<0.0001	169.82 ± 82.68	265.72 ± 122.22 ^a^	<0.0001

All *p*-values are obtained using Student’s *t*-test, and all continuous parameters are reported as mean ± standard deviation. ^a^ The value is significantly (*p* < 0.05) different than that in the control group. PSM, propensity score matching; MMVD, myxomatous mitral valve disease; WBC, white blood cell; NLR, neutrophil-to-lymphocyte ratio; MLR, monocyte-to-lymphocyte ratio; PLR, platelet-to-lymphocyte ratio.

**Table 4 animals-13-02821-t004:** Comparison of clinical and hematological variables between healthy dogs and those with various stages of myxomatous mitral valve disease.

Variables	Control (n = 75)	MMVD B1 (n = 90)	MMVD B2 (n = 49)	MMVD C (n = 92)	MMVD D (n = 18)
Clinical variables					
Age (years)	6.48 ± 4.46	12.21 ± 3.07 ^a^	12.35 ± 2.29 ^a^	12.51 ± 2.95 ^a^	12.06 ± 3.40 ^a^
Hematological variables					
WBC (K/μL)	9.61 ± 5.35	8.32 ± 2.43	9.81 ± 2.77	14.58 ± 5.66 ^a,b,c^	17.07 ± 5.19 ^a,b,c^
Neutrophil (K/μL)	5.46 ± 1.74	5.95 ± 1.65	6.73 ± 1.76^a^	11.14 ± 4.92 ^a,b,c^	13.70 ± 4.82 ^a,b,c,d^
Monocyte (K/μL)	0.48 ± 0.29	0.50 ± 0.26	0.68 ± 0.71	0.99 ± 0.52 ^a,b^	1.11 ± 0.50 ^a,b,c^
Lymphocyte (K/μL)	2.43 ± 1.05	1.85 ± 0.71 ^a^	2.00 ± 0.64 ^a^	1.99 ± 0.60 ^a^	1.74 ± 0.53 ^a^
Platelet (K/μL)	311.58 ± 114.18	400.83 ± 128.56 ^a^	405.39 ± 111.41 ^a^	485.78 ± 152.10 ^a^	504.56 ± 188.81 ^a,b,c^
NLR	2.33 ± 0.93	3.23 ± 0.98 ^a^	3.56 ± 1.08 ^a^	5.71 ± 2.18 ^a,b,c^	8.38 ± 3.22 ^a,b,c,d^
MLR	0.20 ± 0.11	0.28 ± 0.14 ^a^	0.33 ± 0.23 ^a^	0.51 ± 0.23 ^a,b,c^	0.74 ± 0.30 ^a,b,c,d^
PLR	137.61 ± 77.59	234.27 ± 92.46 ^a^	218.32 ± 81.44 ^a^	261.67 ± 104.07 ^a^	326.77 ± 177.01 ^a,b,c^

All *p*-values are obtained using ANOVA, and all continuous parameters are reported as mean ± standard deviation. ^a^ The value is significantly (*p* < 0.05) different than that in the control group; ^b^ The value is significantly (*p* < 0.05) different than that in the MMVD B1 group; ^c^ The value is significantly (*p* < 0.05) different than that in the MMVD B2 group; ^d^ The value is significantly (*p* < 0.05) different than that in the MMVD C group. MMVD, myxomatous mitral valve disease; WBC, white blood cell; NLR, neutrophil-to-lymphocyte ratio; MLR, monocyte-to-lymphocyte ratio; PLR, platelet-to-lymphocyte ratio.

## Data Availability

The data presented in this study are available on request from the corresponding author.

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
