# Peer review of "Prognostic Efficacy of Complete Blood Count Indices for Assessing the Presence and the Progression of Myxomatous Mitral Valve Disease in Dogs"

_animals, 2023, doi:10.3390/ani13182821_

Round 1
Reviewer 1 Report
The article is written in an understandable and interesting way, but it is worth clarifying a few things:
1. Please explain the average size of LA/Ao in the MMVD ACVIM B2 group, by definition this value should be >1.6, and in the study according to Table No. 1 it is 1.45
2. How parameters of inflammation have been linked to heart disease. We know that the disease mainly affects older dogs, which, apart from heart disease, may have other problems, such as inflammation of the mouth due to the presence of tartar, especially small dog breeds. It would be worth making clear that the dogs included in the study did not show clinical signs of inflammation.
Reviewer 2 Report
I have reviewed the manuscript and found it interesting. Its worth publishing this manuscript. I have suggested few changes before the publication of this manuscript
Simple summary
Line 10-21: please provide simple summary in two to three line in such a way that a common man can also easily understand the study.
‘Simple Summary: Neutrophil-to-lymphocyte ratio (NLR), monocyte-to-lymphocyte ratio (MLR), 10 and platelet-to-lymphocyte ratio (PLR) are inflammatory indicators calculated by dividing the ab- 11 solute counts of complete blood count (CBC) indices. These ratios have been studied as potential 12 diagnostic and prognostic biomarkers not only for inflammatory and neoplastic diseases but also 13 for cardiovascular diseases in human medicine. However, research on dogs with cardiac diseases 14 in veterinary medicine is still lacking. The present study compared these indices in healthy dogs 15 and those with myxomatous mitral valve disease (MMVD), the most common cardiac disease in 16 dogs, finding that dogs with MMVD had significantly higher NLR, MLR, and PLR. The indices 17 showed promise as biomarkers for detecting the presence and severity of MMVD. Additionally, 18 higher NLR, MLR, and PLR were associated with reduced survival time. Based on the findings, 19 NLR, MLR, and PLR could serve as cost-effective and readily available diagnostic and prognostic 20 biomarkers for MMVD in dogs’
Abstract
Line 22: don’t start sentence with ‘we’. ‘We investigated………..’
Line 25: on survival time of what ? ‘…………..on survival time’. please mention in the text
Line 26: don’t start sentence with abbreviation for example ‘NLR, MLR, …………..’
Line 30: ‘NLR and MLR were significantly correlated with N-terminal ………..’ don’t start the sentence with abbreviation
Introduction
Line 48: please be specific with disease ‘facilitate disease management.’
Line 48-49: be specific either studies are on human or animals ‘Identifying reliable inflammatory 48 biomarkers in patients with heart failure ….’
Materials and methods
Line 94: please don’t use abbreviation in the headings and subheading ‘Classification of MMVD..’
Line 99: this abbreviation used only once so remove this ‘mitral regurgitation (MR)…’
Line 97-112: I think, in depth details are not required only reference can serve the purpose ‘The MMVD group was further categorized 97 based on the classification system of the American College of Veterinary Internal Medi- 98 cine (ACVIM) [1]. Stage B1 refers to asymptomatic dogs with mitral regurgitation (MR) 99 accompanied by mitral valve pathology, with no radiographic or echocardiographic evi- 100 dence of cardiac remodeling. Stage B2 describes asymptomatic dogs with more advanced 101 MR that is sufficient to cause left atrial (LA) and ventricular enlargement. Enlargement of 102 the left atrium and ventricle was determined based on the following criteria: echocardio- 103 graphic LA: Ao ratio in early diastole ≥1.6, left ventricular internal diameter in diastole 104 (LVIDDN) ≥1.7, and radiographic vertebral heart score (VHS) >10.5. Stage C refers to dogs 105 with clinical signs of left-sided heart failure and a history of tachypnea, respiratory dis- 106 tress, coughing, or syncope. Stage D denotes dogs with end-stage MMVD whose clinical 107 signs are refractory to conventional therapy. If dogs require more than a total dosage of 8 108 mL/kg/day of furosemide, the equivalent dosage of torsemide, or standard doses of med- 109 ications thought to control the symptoms of heart failure (e.g., pimobendan, 0.25–0.3 110 mg/kg PO q 12 h daily), it was considered that the conventional therapy was refractory 111 [1]’
Line 116-117: ‘N-terminal pro-brain natriuretic peptide’ has already been used in introduction. Please don’t try to make abbreviation again here
Line 117: don’t start sentence with abbreviation ‘CBC and automated differential counts were determined within 1 h after………..’
Discussion
Line 465: ‘congestive heart failure’ abbreviation has already been used in introduction section. Please write the abbreviation instead of full term please
Its ok
Author Response
Please see the attachement.
